# Replacing Implicit Regression with Classification in Policy Gradient Reinforcement Learning

## Abstract

Stochastic policy gradient methods are a fundamental class of reinforcement learning algorithms. When using these algorithms for continuous control it is common to parameterize the policy using a Gaussian distribution. In this paper, we show that the policy gradient with Gaussian policies can be viewed as the gradient of a weighted least-squares objective function. That is, policy gradient algorithms are implicitly implementing a form of regression. A number of recent works have shown that reformulating regression problems as classification problems can improve learning. Inspired by these works, we investigate whether replacing this implicit regression with classification can improve the data efficiency and stability of policy learning. Toward this end, we introduce a novel policy gradient surrogate objective for softmax policies over a discretized action space. This surrogate objective uses a form of cross-entropy loss as a replacement for the implicit least-squares loss found in the surrogate loss for Gaussian policies. We extend prior theoretical analysis of this loss to our policy gradient surrogate objective and then provide experiments showing that this novel loss improves the data efficiency of stochastic policy gradient learning in continuous action spaces.

## 1 Introduction

Stochastic policy gradient algorithms are a fundamental class of reinforcement learning (RL) algorithms. In their simplest form, the learning agent runs its current policy to collect data in the form of state, action, and reward transitions to produce a dataset of $(s_i, a_i, \hat{A}_i)$ where $\hat{A}_i$ is an estimate of the *advantage* of taking action $a_i$ in state $s_i$. The learning agent then updates its parameterized and differentiable stochastic policy with a step of gradient ascent on the expected cumulative reward objective. The gradient update increases the log-likelihood of each observed action in proportion to the advantage of that action.

Following Peters and Schaal (2007), we observe that the policy gradient update can be viewed as implicitly optimizing a weighted supervised learning loss function. We particularly focus on the case of continuous control with Gaussian policies in which case we will show that the policy gradient matches the gradient of a weighted least-squares loss function. In this sense, we say that policy gradient algorithms are implicitly implementing (weighted) regression.

A growing body of research (discussed in Section 6) supports the claim that reformulating regression problems as classification problems can boost task performance in supervised regression. Of particular relevance to this work, Imani and White (2018) introduced a form of classification loss for regression problems and showed that it boosts regression accuracy compared to the commonly used squared loss. Subsequently, Farebrother et al. (2024) replaced the squared loss of value-based RL algorithms with this *histogram loss* and found that the approach unlocked new levels of scalability in a wide variety of RL benchmarks. Motivated by these prior works, in this paper, we consider reformulating the implicit regression of stochastic policy gradient RL for continuous action domains as classification. Specifically, this work aims to answer the question:

*Does replacing the least-squares loss and Gaussian policies with a cross-entropy loss and softmax policies over discretized actions improve the data efficiency of policy gradient learning?*

In answering this question, we make the following contributions:

1. We show that the stochastic policy gradient is equal to the gradient of a weighted maximum likelihood objective. In continuous action spaces with Gaussian policies, optimizing this objective amounts to implicit weighted regression with a least-squares loss function.
2. We introduce a novel policy gradient surrogate loss that re-casts the implicit regression of continuous actions as classification of discrete actions.
3. Building on Imani and White (2018), we show that the loss we introduce has a smaller bound on the gradient norm compared to the surrogate loss for Gaussian policies, implying that the new loss is easier to optimize.
4. We empirically investigate the use of cross-entropy losses and softmax policies as an alternative to the widely-used Gaussian policies within stochastic policy gradient algorithms and find that our reformulation leads to increased data-efficiency, more stable learning, and increased final performance.

## 2 PRELIMINARIES

In this section, we introduce RL notation, stochastic policy gradient learning, and the histogram regression loss.

### 2.1 REINFORCEMENT LEARNING

We formalize an RL agent's task environment as a finite-horizon, episodic *Markov decision process* (MDP) with state set $\mathcal{S}$, action set $\mathcal{A}$, transition function, $p : \mathcal{S} \times \mathcal{A} \times \mathcal{S} \to [0, 1]$, reward function $r : \mathcal{S} \times \mathcal{A} \to \mathbb{R}$, discount factor $\gamma$, and initial state distribution $d_0$ (Puterman, 2014). The agent's behavior is determined by its policy, $\pi : \mathbb{S} \times \mathbb{A} \to [0, 1]$, which is a function mapping states to probability distributions over possible actions. Given a policy and task environment, interaction begins at time $t = 0$ in some initial state ($s_0 \sim d_0$) and then proceeds with the agent selecting actions according to its policy ($a_t \sim \pi(\cdot|s_t)$) and the environment responding with a reward, $r_t = r(s_t, a_t)$, and transitioning to a next state ($s_{t+1} \sim p(\cdot|s_t, a_t)$). Interaction continues until the agent reaches a terminal state, at which point, the agent returns to a new initial state and the process begins again. The result of this interaction is a *trajectory*, $h := (s_0, a_0, r_0, s_1, ..., s_T, a_T, r_T)$.

We measure policy performance by the expected discounted return in a given MDP:

$$J(\pi) := \mathbf{E}\left[\sum_{t=0}^{T} \gamma^t R_t | H \sim \pi\right]$$

where $H = (S_0, A_0, R_0, ...S_T, A_T, R_T)$ is a random variable representing a trajectory and $H \sim \pi$ denotes sampling $H$ by running $\pi$ for one episode. In RL, the transition and reward functions of the task MDP are unknown. RL algorithms are designed to collect trajectory data from the task MDP and use this data to return a policy, $\pi^* \in \arg\max_\pi J(\pi)$.

### 2.2 POLICY GRADIENT REINFORCEMENT LEARNING

In policy gradient reinforcement learning, the agent's policy is parameterized by a vector, $\boldsymbol{\theta}$, and the policy is differentiable with respect to these parameters. Policy gradient RL algorithms optimize the policy through gradient ascent over $\boldsymbol{\theta}$ to maximize $J(\pi_{\boldsymbol{\theta}})$. The gradient of the $J(\boldsymbol{\theta})$ with respect to $\boldsymbol{\theta}$, or *policy gradient*, is typically expressed as:

$$\nabla_{\boldsymbol{\theta}} J(\pi_{\boldsymbol{\theta}}) \propto \mathbb{E}_{\boldsymbol{s} \sim d_{\pi_{\boldsymbol{\theta}}}, \boldsymbol{a} \sim \pi_{\boldsymbol{\theta}}(\cdot|\boldsymbol{s})} \left[ A^{\pi_{\boldsymbol{\theta}}}(\boldsymbol{s}, \boldsymbol{a}) \nabla_{\boldsymbol{\theta}} \log \pi_{\boldsymbol{\theta}}(\boldsymbol{a}|\boldsymbol{s}) \right], \tag{1}$$

where $A^{\pi_{\boldsymbol{\theta}}}(\boldsymbol{s}, \boldsymbol{a})$ is the *advantage* of choosing action $\boldsymbol{a}$ in state $\boldsymbol{s}$ and quantifies the extra expected reward that will be obtained when taking $\boldsymbol{a}$ instead of sampling an action from $\pi_{\boldsymbol{\theta}}$ in $\boldsymbol{s}$, and $d_{\pi_{\boldsymbol{\theta}}} : \mathbb{S} \to [0, 1]$ is the expected distribution of states that will be seen when running $\pi_{\boldsymbol{\theta}}$ in the task MDP (Schulman et al., 2016). Note that in reality Equation (1) is *not* the gradient of $J(\pi_{\boldsymbol{\theta}})$ but is a widely used and biased approximation of it (Thomas, 2014; Nota and Thomas, 2020).

### 2.3 SUPERVISED REGRESSION AS CLASSIFICATION

In supervised learning, regression problems are typically formulated using the least-squares loss function:

$$\mathcal{L}_{\text{LS}}(\boldsymbol{\theta}) := \frac{1}{m} \sum_{j=1}^{m} ||h_{\boldsymbol{\theta}}(x_j) - \boldsymbol{y}_j||_2^2, \tag{2}$$

for predictor $h_{\boldsymbol{\theta}} : \mathcal{X} \rightarrow \mathbb{R}^d$ that maps inputs $x \in \mathcal{X}$ to labels $\boldsymbol{y} \in \mathbb{R}^d$ and $m$ is the number of training examples. Though minimizing the squared distance to a desired target is a natural choice of the loss function, an alternative is to discretize the label space and reformulate regression as classification with a cross-entropy loss function. Perhaps counterintuitively, this reformulation has been shown to be beneficial in practice (Farebrother et al., 2024) and theory (Imani et al., 2024).

For exposition, in this section, we will only consider the case when $d = 1$. Let $y_{\min}$ and $y_{\max}$ be a minimum and maximum bound on the predicted value from $h_{\boldsymbol{\theta}}(x)$. Since classification requires a discrete label set, we discretize the interval $[y_{\min}, y_{\max}]$ uniformly into $k$ bins and the predictor $h_\theta(x)$ outputs $k$ logits that parameterize a softmax distribution over the $k$ bins. Let $\tilde{y}_i$ be the center of the $i^{\text{th}}$ bin and $\hat{p}_i(x)$ be the probability of the $i^{\text{th}}$ bin output by $h_{\boldsymbol{\theta}}(x)$. The scalar-valued prediction for $y$ given $x$ is the expected value of $\tilde{y}_i$ under $\hat{p}(x)$ or $\sum_{i=1}^{k} \tilde{y}_i \hat{p}_i(x)$.

We train $h_\theta$ using a cross-entropy loss between $h_{\boldsymbol{\theta}}(x)$ and a target distribution that is specified from $y$. Following the notation of Imani and White (2018), we denote this target distribution as $q_y$. We then train $h_{\boldsymbol{\theta}}$ by minimizing the cross-entropy loss:

$$\mathcal{L}_{\texttt{CE}}(\boldsymbol{\theta}) := \frac{1}{m} \sum_{j=1}^{m} \sum_{i}^{k} q_{y_j}(i) \log \hat{p}_i(x_j). \tag{3}$$

In this work, we will consider two choices for the target distribution $q_y$. A straightforward choice is a 1-hot distribution with the bin corresponding to $y$ receiving probability 1. However, this choice potentially discards information about the spatial structure and ordinality of the label space that the least-squares loss preserves. Imani and White (2018) and Farebrother et al. (2024) found that a histogram approximation to a Gaussian distribution with a mean of $y$ and the standard deviation chosen as a hyper-parameter was a better choice for this reason (see Figure 1 in (Imani et al., 2024) for illustration). Using a histogram approximation of a Gaussian in Equation (3) results in a loss that Imani and White (2018) called HL-Gauss. Optimizing HL-Gauss for input $x_j$ increases the probability of outputting the target label $y_j$ the most while also increasing the probability of values close to $y_j$.

## 3 IMPLICIT REGRESSION IN POLICY GRADIENT RL

In this section, we show how policy gradient RL updates with Gaussian policies are equivalent to weighted regression updates. First, we note that policy gradient algorithms are often implemented to maximize the surrogate loss:

$$\mathcal{L}_{\texttt{surr}}(\boldsymbol{\theta}) := \frac{1}{m} \sum_{j=1}^{m} A_{\pi_{\boldsymbol{\theta}}}(\boldsymbol{s}_j, \boldsymbol{a}_j) \log \pi_{\boldsymbol{\theta}}(\boldsymbol{a}_j | \boldsymbol{s}_j), \tag{4}$$

where we have obtained $m$ state-action pairs by running some policy. Assuming that the data was collected on-policy (i.e., by running $\pi_{\boldsymbol{\theta}}$) then the gradient of Equation (4) is an unbiased estimator of Equation (1) (Foerster et al., 2018). We interpret $\mathcal{L}_{\texttt{surr}}$ as a weighted supervised learning loss with states as inputs, actions as labels, and the weight on each sample given by the advantage function.

When the task MDP has continuous actions, the most common policy parameterization is a multivariate Gaussian where the mean and covariance are given as functions of the state that are parameterized by $\boldsymbol{\theta}$. That is $\pi_{\boldsymbol{\theta}}(\boldsymbol{a}|\boldsymbol{s}) = \mathcal{N}(\boldsymbol{a}; \mu_{\boldsymbol{\theta}}(\boldsymbol{s}), \Sigma_{\boldsymbol{\theta}}(\boldsymbol{s}))$. For the sake of exposition, we will treat $\Sigma_{\boldsymbol{\theta}}(s)$ as a constant identity matrix and focus on $\mu_{\boldsymbol{\theta}}(s)$.[1] Under a Gaussian parameterization, the surrogate objective becomes a weighted least-squares regression problem:

$$\arg\max_{\boldsymbol{\theta}} \mathcal{L}_{\texttt{surr}}(\boldsymbol{\theta}) = \arg\min_{\boldsymbol{\theta}} \mathcal{L}_{\texttt{PG-LS}}(\boldsymbol{\theta}) := \frac{1}{m} \sum_{j=1}^{m} A_{\pi_{\boldsymbol{\theta}}}(\boldsymbol{s}_j, \boldsymbol{a}_j) \frac{1}{2} ||\boldsymbol{a}_j - \mu_\theta(\boldsymbol{s}_j)||_2^2 + \texttt{const} \tag{5}$$

It can now be seen that the policy gradient surrogate loss for Gaussian policies is a weighted least-squares problem that resembles Equation (2).

---

[1] In our experiments, we will learn a state-independent covariance matrix when considering Gaussian policies. A non-identity covariance matrix means that the policy gradient method is implicitly implementing heteroscedastic regression.

The connection between policy optimization and supervised learning has been previously made by Peters and Schaal (2007); Peng et al. (2019); Abdolmaleki et al. (2018) in the context of formulating policy optimization with KL-divergence constraints. Under the formulation in these past works, policy optimization can also be cast as weighted regression. The key difference between these works and ours is that the KL-divergence constraint results in the weighting function being $\exp(\frac{1}{\tau} A_{\pi_{\boldsymbol{\theta}}}(\boldsymbol{s}, \boldsymbol{a}))$ rather than $A_{\pi_{\boldsymbol{\theta}}}(\boldsymbol{s}, \boldsymbol{a})$.[2]

# 4 Replacing Implicit Regression with Classification

Empirical evidence in supervised learning and RL suggests there is an empirical benefit to reformulating regression problems as classification problems. Our goal in this work is to understand if this benefit translates to policy gradient learning if we reformulate the implicit regression in policy gradient methods as classification. Toward this goal, we first describe how we can represent continuous action policies as policies over discrete actions and then introduce a new policy gradient surrogate objective for training these policies.

## 4.1 Policy Representation

To recast regression as classification, we first need to parameterize the policy we are learning as a distribution over a finite set, $\mathcal{Z}$, where each continuous $\boldsymbol{a} \in \mathcal{A}$ maps to an element of $z \in \mathcal{Z}$. The naive way to accomplish this mapping is to discretize the continuous space using a multi-dimensional grid with $k$ bins along each dimension. The grid representation is useful in that it can learn policies in which different action dimensions are correlated. The downside of this representation is that the number of discrete actions will be exponential in the native action space dimensionality.

To make discrete action policies tractable, we make the simplifying assumption that each action dimension is selected independently of the others. This assumption is reasonable as it is already standard practice when using Gaussian policies to use a diagonal covariance matrix. Thus, in comparison to such Gaussian policies, the policy representation that we introduce is only limited in terms of the granularity of the discretization. This simplification means that after discretization, each dimension has $k$ bins and the policy network only needs to output $kd$ values instead of $k^d$. The limitation of this assumption is that the space of possible distributions over actions that the policy can represent is now limited.

Formally, we learn a policy that outputs $d$ softmax distributions (one for each action dimension) where each distribution is over a finite set $\mathcal{Z}_l$ for $l \in \{1, ..., d\}$. We map each continuous value in $[a_{\min}, a_{\max}]$ to an element of $\mathcal{Z}_l$. In this work, we use a uniform discretization with $k$ bins and let $c := \frac{a_{\max} - a_{\min}}{k}$ be the width of each bin. The elements of $\mathcal{Z}_l$ form an ordered set where the $i^{\text{th}}$ element, $z_l^i$, represents the range $[a_{\min} + c \cdot (i-1), a_{\min} + c \cdot i]$ for $i \in \{1, ...k\}$. Let $a_l^i$ be the center of this range. We denote $\pi_{\boldsymbol{\theta}}^l(\cdot|\boldsymbol{s})$ as the policy distribution over action dimension $l$. To sample from this policy representation, we first sample $z_l^i \sim \pi_{\boldsymbol{\theta}}^l(\cdot|\boldsymbol{s})$ for each dimension $l$. We then return the associated $a_l^i$ as the value of the action for that dimension.[3]

## 4.2 Policy Gradient Learning as Classification

Now, to replace the implicit regression in Equation (5) with classification, we replace the least-squares portion of $\mathcal{L}_{\text{PG}-\text{LS}}$ with a cross-entropy loss. By doing so, we obtain the loss function:

$$\mathcal{L}_{\text{PG}-\text{CE}}(\boldsymbol{\theta}) := \frac{1}{m} \sum_{j=1}^{m} A_{\pi_{\boldsymbol{\theta}}}(\boldsymbol{s}_j, \boldsymbol{a}_j) \sum_{l=1}^{d} \sum_{i=1}^{k} q_{a_{j,l}}(i) \log \pi(\tilde{a}_l^i|\boldsymbol{s}), \quad (6)$$

where $q_{a_l}$ is a target probability distribution over action dimension $l$ that is defined in terms of the sampled action $a_{j,l}$. We consider two choices for the target distribution: the 1-hot distribution that places all probability mass on the observed action and a histogram approximation to a Gaussian

---

[2]We informally experimented with using the exponentiated advantage at the start of our investigation. We found that the non-exponentiated advantage gave better results and had fewer hyper-parameters to tune.

[3]We choose to deterministically return the center of the range for simplicity but alternative choices could be made. For instance, we could uniformly sample from the range.

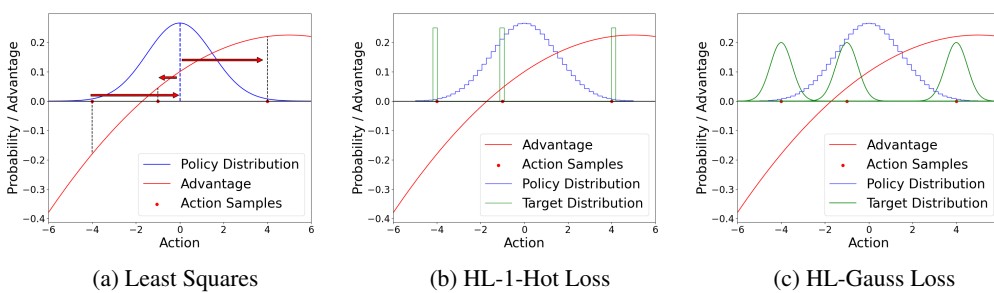

(a) Least Squares   (b) HL-1-Hot Loss   (c) HL-Gauss Loss

Figure 1: Illustrations of the three loss functions that we consider in a single-state problem where three 1-d actions have been sampled to update the policy.

distribution centered at dimension $l$ of action $\boldsymbol{a}$. We call these two instantiations of our new loss HL-1-Hot, and HL-Gauss, respectively, following prior work in supervised learning Imani and White (2018). For the latter, the standard deviation, $\sigma$, of the approximated Gaussian is a method hyperparameter; we follow Farebrother et al. (2024) by tuning $\eta := \frac{\sigma}{c}$ instead of $\sigma$ directly.

Figure 1 illustrates policy optimization with each of the three losses we consider. When learning with Gaussian policies, the policy gradient moves the mean of the policy's distribution toward the action samples in the same way that regression moves the mean prediction toward target values. For policy gradient learning, this movement is made in proportion to the advantage of each action; negative advantages repel the movement. When learning with softmax policies and the HL-1-Hot loss, the update increases the probability of only the sampled actions in proportion to their advantages; negative advantages lead to decreased probability. When using the HL-Gauss loss, the update moves the policy distribution toward target distributions centered on sampled actions with positive advantages.

### 4.3 BOUNDS ON THE NORMS OF THE LOSS GRADIENTS

Imani and White (2018) found that stable gradients were a benefit of the HL-Gauss loss compared to either a 1-hot cross-entropy loss or a least-squares loss for regression. Here, increased stability means that the norm of the loss gradient has a smaller upper bound compared to the gradient of $\mathcal{L}_{\text{PG-LS}}$. Imani and White (2018) attribute the utility of a small gradient norm to prior theoretical work (Hardt et al., 2016) showing that a loss with a small Lipschitz constant provides an improved upper bound on generalization performance in supervised learning. Furthermore, a smaller bound on the gradient norm may ease the difficulty in setting a learning rate for stable, consistent learning progress. In this section, we extend this analysis to the policy gradient surrogate losses that we consider in this paper and show that the gradient of $\mathcal{L}_{\text{PG-CE}}$ has a smaller norm compared to $\mathcal{L}_{\text{PG-LS}}$.

For conciseness, we will only consider the case that $d = 1$. In our analysis, we assume that the policy function is represented by a neural network. Let $f_{\boldsymbol{\phi}}(\boldsymbol{s})$ denote the feature representation of $\boldsymbol{s}$ at the penultimate layer of the network and $\boldsymbol{\phi}$ is the network parameters before the final layer. Define the policy gradient cross-entropy loss at a given state-action pair to be $\mathcal{L}_{\text{PG-CE}}(\boldsymbol{\theta}, \boldsymbol{s}, \boldsymbol{a}) :=$ $A_{\pi_{\boldsymbol{\theta}}}(\boldsymbol{s}, \boldsymbol{a}) \sum_{i=1}^{k} q_{\boldsymbol{a},i} \log \hat{p}_i(\boldsymbol{s})$. For our analysis, we define $\boldsymbol{\theta} := [\boldsymbol{w}_1, .., \boldsymbol{w}_k, \boldsymbol{\phi}]$ where $\boldsymbol{w}_i$ is the weight vector for the inputs of output logit $i$. We further assume that the policy network's output is $l$-Lipschitz, i.e $\|\nabla_{\boldsymbol{\phi}} \boldsymbol{w}_j^\top f_{\boldsymbol{\phi}}(\boldsymbol{s})\| \leq l$ with respect to $\boldsymbol{\phi}$. We can then bound the norm of the policy gradient cross-entropy loss as follows.

**Proposition 1.** *The norm of $\nabla_{\boldsymbol{\theta}} \mathcal{L}_{\text{PG-CE}}$ is upper-bounded as follows:*

$$\|\nabla_{\boldsymbol{\theta}} \mathcal{L}_{\text{PG-CE}}(\boldsymbol{\theta}, \boldsymbol{s}, \boldsymbol{a})\| \leq \left| A_{\pi_{\boldsymbol{\theta}}}(\boldsymbol{s}, \boldsymbol{a}) \right| \left( l + \|f_{\boldsymbol{\phi}}(\boldsymbol{s})\| \right) \left( \sum_{i=1}^{k} |q_{\boldsymbol{a},i} - \pi_{\boldsymbol{\theta}}(a_i|\boldsymbol{s})| \right) \tag{7}$$

*Proof.* See Appendix A.             $\square$

In comparison to the supervised learning setting of Imani et al. (2024), our bound on the gradient norm at any state-action pair is multiplied by the magnitude of the advantage, which is ex-

pected as $\mathcal{L}_{\text{PG}-\text{CE}}$ is equal to $\mathcal{L}_{\text{CE}}$ multiplied by the advantage. The next question is how this bound compares to the upper bound on the gradient norm of $\mathcal{L}_{\text{PG}-\text{LS}}$. We define $\mathcal{L}_{\text{PG}-\text{LS}}(\boldsymbol{\theta}, \boldsymbol{s}, \boldsymbol{a}) :=$ $A_{\pi_{\boldsymbol{\theta}}}(\boldsymbol{s}, \boldsymbol{a})\frac{(\boldsymbol{a} - \boldsymbol{w}^\top f_{\boldsymbol{\phi}}(\boldsymbol{s}))^2}{2\sigma}$. For Gaussian policies, we define $\boldsymbol{\theta} := [\boldsymbol{w}, \boldsymbol{\phi}]$ where $\boldsymbol{w}$ is the weights of the final linear layer. The mean of the Gaussian policy at state $\boldsymbol{s}$ is equal to $\boldsymbol{w}^\top f_{\boldsymbol{\phi}}(\boldsymbol{s})$. As we did for the logits of softmax policies, we assume that the mean is $l$-Lipschitz i.e $\|\nabla_{\boldsymbol{\phi}}\boldsymbol{w}^\top f_{\boldsymbol{\phi}}(\boldsymbol{s})\| \leq l$ with respect to $\boldsymbol{\phi}$. We then obtain the following bound on the norm of the policy gradient for Gaussian policies.

**Proposition 2.** *The norm of $\nabla_{\boldsymbol{\theta}}\mathcal{L}_{\text{PG}-\text{LS}}$ is upper-bounded as follows:*

$$\|\nabla_{\boldsymbol{\theta}}\mathcal{L}_{\text{PG}-\text{LS}}(\boldsymbol{\theta}, \boldsymbol{s}, \boldsymbol{a})\| \leq \frac{1}{\sigma}\left|A_{\pi_{\boldsymbol{\theta}}}(\boldsymbol{s}, \boldsymbol{a})\right|\left(\left(l + \|f_{\boldsymbol{\phi}}(\boldsymbol{s})\|\right)\right)\left|\boldsymbol{a} - \boldsymbol{w}^\top f_{\boldsymbol{\phi}}(\boldsymbol{s})\right| \tag{8}$$

*Proof.* See Appendix B. □

The bound in Proposition 2 has two different factors from the bound in Proposition 1. The factor $\frac{1}{\sigma}$ may decrease the bound early in training when $\sigma$ is large. However, as $\sigma \to 0$, this factor causes the gradient norm to explode. In fact, in our experiments (and other works in the literature), we found it necessary to either normalize the gradient or clip $\sigma$ above $0$ to enable stable learning. The final factor in each of these bounds is difficult to compare but will generally be of similar magnitude. The term $\sum_{i=1}^k |q_{\boldsymbol{a},i} - \pi_{\boldsymbol{\theta}}(a_i|\boldsymbol{s})|$ in the cross-entropy loss is always bounded by 2 whereas the term $|\boldsymbol{a} - \boldsymbol{w}^{\phi(s)}|$ is bounded by the range of the action-space which can be re-scaled to a comparable value. This analysis shows that replacing implicit regression with classification leads to a policy gradient surrogate loss with a smaller bound on the gradient norm with less variation during learning.

## 5 EMPIRICAL ANALYSIS

We next conduct an empirical study designed to answer the following questions:

1. Does replacing the implicit regression of the policy gradient surrogate loss with a cross-entropy loss increase the data efficiency of stochastic policy gradient methods?
2. Are observed increases in learning efficiency due to improved exploration or optimization?

We also test the sensitivity of different loss functions to noise in advantage estimates, action space dimensionality, and hyper-parameter sensitivity.

### 5.1 EMPIRICAL SET-UP

To investigate these questions, we run learning trials in the following continuous action testbed domains: a continuous action bandit environment, linear quadratic regulator (LQR), continuous Acrobot, continuous Mountain Car, HalfCheetah, and Ant (Towers et al., 2023). Please refer to Appendix C.1 for detailed descriptions of each environment.

For simplicity, we use a stochastic actor-critic algorithm as the base policy gradient algorithm (Sutton and Barto, 2018). We use $n$-step returns to estimate the advantage function, the Adam optimizer (Kingma and Ba, 2015), and clip gradients during training. For advantage estimation, we fit a state-dependent value function using an MSE loss with observed returns as targets.

We compare the following policy representations and loss functions:

1. **Gaussian policies with least-squares loss.** The policy is a neural network that predicts a mean action given the current state and has state-independent diagonal covariance that we parameterize as the log standard deviation. Actions are sampled from a Gaussian distribution parameterized by this mean and covariance. We use the standard policy gradient surrogate loss to train this policy.
2. **Softmax policies with HL-Gauss loss.** This method uses the policy representation described in Section 4.1. We use Equation (6) as the surrogate loss to train this policy with the target distribution a Gaussian. We tune the learning rate, $k$, and $\eta$.

3. **Softmax policies with 1-hot loss.** The policy is parameterized the same as when using HL-Gauss. Instead of the HL-Gauss loss, we instead use a target distribution that is 1-hot on the bin for the action that was taken. This method tests whether it is important to preserve the spatial structure of the action space in the loss function.

We refer the reader to Appendix C.2 for training details of all the algorithms such hyperparameters, batch sizes, policy architectures, etc. The best hyper-parameters for each method are chosen using a sweep where the best is chosen based on the highest average undiscounted return (averaged across all trials) achieved at the end of training. The sensitivity of each method is discussed in Appendix C.3. We generally find that the cross-entropy losses are more robust to hyper-parameters (Appendix C.4 gives performance profiles for each method across all hyper-parameters tested).

## 5.2 EMPIRICAL RESULTS

We now present the results of our empirical study.

### 5.2.1 COMPARATIVE RESULTS

We present our main results of evaluating the performance of the classification vs. regression loss in Figure 2 on several environments. In general, we find that re-casting the regression loss as a cross-entropy loss significantly boosts learning efficiency. An exception was in the Acrobot environment (shown in Figure 2b), where regression outperformed classification methods. In almost all instances, however, we observe that agents that minimize a cross-entropy loss learn faster and achieve a higher return at the end of the training period.

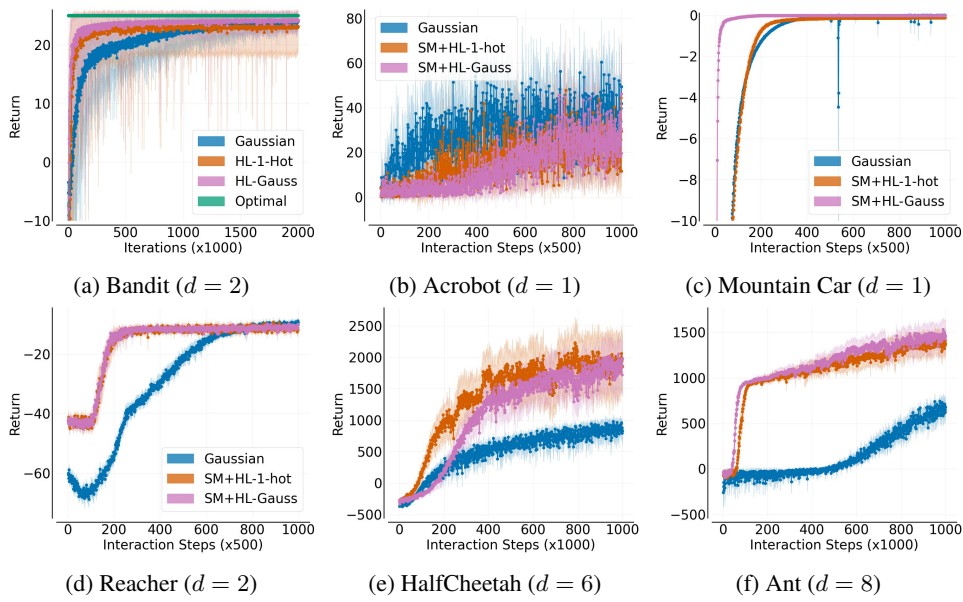

Figure 2: Highest undiscounted training returns achieved by each algorithm as a function of environment interaction steps after a hyperparameter sweep. SM is a softmax policy and HL is the histogram loss. Results are the mean averaged over 20 trials and the shaded region represents the 95% confidence interval. Higher is better. The optimal return can be computed exactly in the Bandit and LQR settings. For each domain, we also give the action-dimensionality, $d$.

### 5.2.2 SENSITIVITY TO ADVANTAGE NOISE AND ACTION DIMENSIONALITY

This subsection examines the sensitivity of different methods to noise in the advantage function estimate as well as the dimensionality of the agent's action space. We use the stateless continuous bandit domain for these experiments and keep all hyper-parameters fixed at their default values that were tuned for the experiments in the preceding subsection. In this domain, the reward is $r(\boldsymbol{a}) \leftarrow 25 - \frac{1}{d}(\boldsymbol{a} - \boldsymbol{5})^2 + \epsilon$ where $\epsilon \sim \mathcal{N}(0, \sigma)$ and the default values of $d$ and $\sigma$ are 2 and 1

respectively. To determine sensitivity to advantage noise and dimensionality, we independently vary the standard deviation, $\sigma$, of the reward and the dimensionality, respectively.

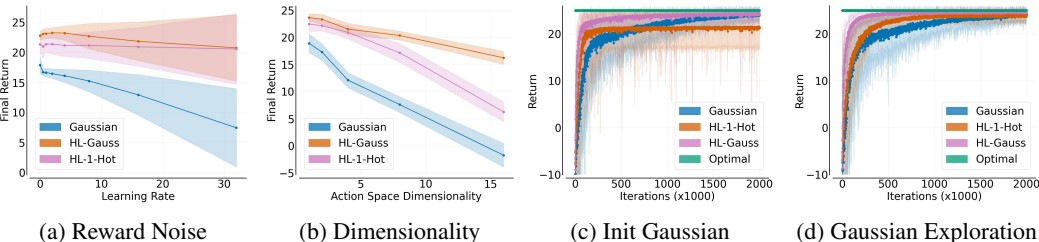

| (a) Reward Noise | (b) Dimensionality | (c) Init Gaussian | (d) Gaussian Exploration |

Figure 3: Environment sensitivity and exploration ablation studies. Error bars give a 95% confidence interval over 50 trials.

Figure 3a shows that both cross-entropy loss methods are more insensitive to reward noise, whereas Gaussian policies are strongly affected by it. Figure 3b shows that all methods degrade in performance as the action space dimensionality increases. However, the cross-entropy loss methods maintain a higher level of final performance for all tested dimensionalities.

### 5.2.3 EXPLORATION VS OPTIMIZATION

The use of histogram losses for policy gradient learning is qualitatively different than past studies on replacing regression with classification because the softmax representation does not just affect policy optimization but also affects the data distribution of the learning agent. This observation motivates our second empirical question as to whether the observed benefits arise from improved exploration, improved optimization, or both. We note that prior work on using discretized action spaces for continuous control has hypothesized that the benefit is entirely due to improved exploration (Tang and Agrawal, 2019; OpenAI et al., 2019).

To answer our question, we use the Bandit domain and repeat our main experiment under two additional conditions.

1. **Init Gaussian.** We initialize softmax policies to approximate the same initial Gaussian distribution that Gaussian policies use.

2. **Gaussian Exploration** At each iteration, we transform the softmax policy into a Gaussian policy and sample actions from this policy for exploration. To do so, we compute the mean and variance of bin centers under the softmax distribution. This mean and variance then parameterize the Gaussian exploration distribution.

The first condition is intended to control for the potential of wider initial exploration under a uniform softmax distribution and the second condition is intended to control for the potential of more flexible exploration. Figure 3c and Figure 3d show learning curves under **Init Gaussian** and **Gaussian Exploration** respectively. We contrast these figures with Figure 2a which shows learning curves when we initialize softmax distributions as uniform distributions. No significant negative effect for HL-1-Hot and HL-Gauss is observed under Gaussian initialization. This finding suggests that improved initial exploration with a softmax policy is not a reason for the increase in data efficiency that we have observed. With **Gaussian Exploration**, we observe that both HL-1-Hot and HL-Gauss *improve* their rate of convergence, indicating that softmax exploration is *not* the key reason for increased data efficiency in this domain. For HL-1-Hot, the use of Gaussian exploration removes the sub-optimal convergence we observe for this method in the base setting. We suspect that this result is due to the fact that the Gaussian exploration adds information about how close different actions are to one another.

Our main takeaway from this experiment is that the cross-entropy losses improve the optimization properties of policy gradient learning. These findings align with our theoretical analysis in Section 4.3 that shows a smaller norm on the gradient of the cross-entropy losses. Our findings do *not* indicate that exploration cannot also be a benefit of discretization in some RL domains as prior work has conjectured (OpenAI et al., 2019; Tang and Agrawal, 2019).

## 6 RELATED WORK

Several related ideas to the contributions of our paper have been previously explored in the literature.

**Policy Search as Supervised Learning**    We have shown how stochastic policy gradient algorithms with Gaussian policy representations can be viewed as implicitly solving a regression problem. A number of prior works have tried to re-cast RL as supervised learning. Peters and Schaal (2007) derive the reward-weighted regression algorithm in order to cast policy search in continuous control spaces as a weighted regression problem. Recent works such as MPO (Abdolmaleki et al., 2018), advantage-weighted regression (Peng et al., 2019), and advantage-weighted actor-critic (Nair et al., 2021) use similar derivations to develop policy search methods that implicitly solve least-squares optimization problems when using Gaussian policies. An alternative approach to RL as SL is upside-down RL that proposes to learn a policy $\pi(s, g)$ by regressing state-return pairs to the action taken in state $s$ before ultimately receiving return $g$. The optimal action for state $s$ is then predicted as $\pi(s, g^\star)$ (Schmidhuber, 2020). Upside-down RL is also the basis for the decision-transformer approach to offline and online RL (Chen et al., 2021). Our study is complementary to these prior works in its focus on recasting implicit regression as classification in policy search.

**Recasting Regression as Classification**    In the supervised learning literature, several works have studied the empirical and theoretical benefits of replacing the least-squares regression loss with a cross-entropy classification loss. Our approach most closely follows the approach of Imani and White (2018); Imani et al. (2024) due to our use of the HL-Gauss loss as a means to preserve the spatial structure of the action space. Zhang et al. (2023) found that the cross-entropy loss encouraged better representations in regression problems and Pintea et al. (2023) found that casting regression as classification helped with class imbalances. While these findings are focused on the supervised-learning case and 1-hot cross-entropy losses, it would be interesting to see whether similar benefits can be found in policy gradient learning. Lastly, we note that a number of applied works, primarily in computer vision, have found classification formulations of regression to give superior empirical performance (Cao et al., 2018; Kendall et al., 2017; Li et al., 2022; Rothe et al., 2015, e.g.,). The policy gradient learning setting is quite different from supervised learning as the function we are learning also directly determines the data distribution being learned over.

**Alternatives to Gaussian Policies in Continuous Action Domains**    In order to recast continuous action policy gradient learning as a classification problem, we discretized each dimension of the action-space and learned softmax policies over each dimension. Tang and Agrawal (2019) previously found discretization to be effective in continuous control benchmarks and credited the benefit to improved exploration. OpenAI et al. (2019) also conjectured improved exploration to be a reason to prefer discretized actions. Our theoretical and empirical analysis complements these works by emphasizing that the cross-entropy loss itself is desirable for learning. As mentioned in the introduction, naive discretization can lead to an exponentially sized action space that would make it intractable to represent the policy. One prior work has addressed this increase in the size of the action space by sequentially selecting the discretized action for each dimension (Metz et al., 2019). Though Metz et al. (2019) introduced this approach to enable q-learning (Watkins and Dayan, 1992) in continuous action domains, it would be interesting to consider new policy parameterizations based on this approach that could be trained with classification losses. Alternative policy distributions such as truncated Gaussians (Fujita and Maeda, 2018), Beta distributions (Chou et al., 2017), and non-parametric distributions Tessler et al. (2019) have also been explored as alternatives to the Gaussian representation. Continuous actions can be directly sampled from these distributions and it would be interesting to investigate if our findings pertain in some form to these alternative representations.

## 7 DISCUSSION AND LIMITATIONS

We have found that the cross-entropy-based policy gradient surrogate loss that we introduced in this work generally leads to more data-efficient policy gradient learning across the continuous control domains where we evaluated it. Results showed that performance improved even as the action-space dimensionality increased which shows the viability of simply selecting each action dimension independently. These results suggest that reformulating the weighted regression found in policy gradient learning for Gaussian policies as weighted classification for softmax policies can be an effective strategy in continuous control RL applications.

We found that the performance difference between the HL-1-Hot and HL-Gauss surrogate losses was often small and the ordering of the methods changed across domains. This result was somewhat surprising to us as HL-1-Hot discards spatial information about the action-space when reinforcing actions whereas HL-Gauss makes use of this information through the Gaussian target distribution. We do find in some cases that HL-1-Hot may be more prone to find local optima (i.e., it converges to a discretized action that is adjacent to the optimal discretized action), however, the loss in performance from these cases is small in the benchmarks we considered.

Perhaps the principle limitation of the HL-Gauss loss is the need to discretize the action space so that a softmax policy can be used. The result is that the true deterministic, optimal policy may not be representable, e.g., if the optimal action in some state is not a bin center. The degree of this limitation depends upon the properties of a domain and how necessary it is for the policy to output precise actions for acceptable performance. In our experiments, we observed mildly adverse effects from discretizing the action-space only in limited number of settings such as in Figure 7b, but did not observe negative effects in general when evaluated on 7 environments. This result could be because the domains we considered do not require precise control for high return. It could also indicate that learning with Gaussian policies is sufficiently slow that we never reach the point where their improved representation power becomes useful. Further small-scale studies on carefully controlled toy problems could help understand when discretization is *not* a viable strategy.

In terms of broader societal impacts, RL algorithms have the potential for both positive and negative impacts depending on the application. This paper studies fundamental RL algorithms using theoretical and empirical analysis in toy problems and benchmarks rather than specific applications. Hence, its immediate societal impact is neutral.

# 8 CONCLUSION AND FUTURE WORK

This paper has studied the degree to which stochastic policy gradient algorithms can be improved for continuous action domains by replacing an implicit least-squares loss term with a cross-entropy loss term in the policy gradient surrogate objective. We first derived the connection between the policy gradient for Gaussian policies and a certain weighted least-squares optimization problem. We then introduced a novel loss function that replaces the implicit weighted regression loss for Gaussian policies with a weighted cross-entropy loss for softmax policies. We showed theoretically that this loss enjoys a smaller bound on gradient norms and then showed empirically that this novel loss improves the data efficiency of a prototypical stochastic actor-critic method for continuous control.

This paper raises new directions for future research. First, for simplicity, our empirical study focused on a prototypical actor-critic method and left open the question of how histogram losses might be integrated into more advanced policy gradient methods. For methods such as MPO (Abdolmaleki et al., 2018), the application of our novel loss is immediate. For methods such as PPO or TRPO, this integration requires creating a novel surrogate loss that can be optimized for multiple steps whereas this paper has only considered a single step of gradient descent. Second, prior work in discrete action spaces has noted limitations of softmax policies for non-stationary problems and proposed alternative policy update schemes Garg et al. (2022). It would be interesting to study whether these losses show improvement over the Gaussian policy surrogate loss in non-stationary, continuous control problems.

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

## A  PROOF OF PROPOSITION 1

**Proposition 1.** *The norm of $\nabla_\theta \mathcal{L}_{\text{PG-CE}}$ is upper-bounded as follows:*

$$\|\nabla_\theta \mathcal{L}_{\text{PG-CE}}(\theta, s, a)\| \leq \left| A_{\pi_\theta}(s, a) \right| \left( l + \|f_\phi(s)\| \right) \left( \sum_{i=1}^{k} |q_{a,i} - \pi_\theta(a_i|s)| \right) \tag{7}$$

*Proof.* The proof for this proposition follows largely from Imani and White (2018) with the key difference being that the advantage estimate for a given state-action pair is weighted by $A_\pi(s, a)$. However, we list out the proof for completeness for the reader.

Let us represent $\theta = [w_1, ..., w_k, \phi,]^T$ where $\phi$ is network parameters up to and including the penultimate layer and $w_i$ is the weights in the final layer that are associated with output $i$. The unnormalized softmax logit for the $i^{th}$ bin is given as $b_i = e^{f_\phi(s)^T w_i}$. Then $\forall j \neq i$ and $j \in \{1, 2, \ldots k\}$,

$$\frac{\partial}{\partial b_i} \pi_\theta(a_j|s) = \frac{\partial}{\partial b_i} \frac{e_j}{\sum_{l=1}^k e_l} = -\frac{e_j}{\left(\sum_{l=1}^k e_l\right)^2} e_i = -\pi_\theta(a_j|s)\pi_\theta(a_i|s) \tag{9}$$

Similarly, for $j = i$, we can write,

$$\frac{\partial}{\partial b_i} \pi_\theta(a_j|s) = \frac{e_i}{\sum_{l=1}^k e_l} - \frac{e_i}{\left(\sum_{l=1}^k e_l\right)^2} e_i = \pi_\theta(a_i|s)\left[1 - \pi_\theta(a_i|s)\right] \tag{10}$$

Using the above expressions we can compute the partial derivative of the histogram loss *without the advantage weighting*:

$$\frac{\partial}{\partial b_i} \sum_{j=1}^k q_{a_l,j} \log \pi_\theta(a_j|s) = \sum_{j=1,j\neq i}^k \frac{-q_{a_l,j}}{\pi_\theta(a_i|s)} \pi_\theta(a_j|s)\pi_\theta(a_i|s) + \frac{q_{a_l,i}}{\pi_\theta(a_i|s)} \pi_\theta(a_i|s)\left[1 - \pi_\theta(a_i|s)\right] \tag{11}$$

$$= q_{a_l,i} - \pi_\theta(a_i|s) \sum_{j=1,j\neq i}^k q_{a_l,j} - q_{a_l,i}\pi_\theta(a_i|s) \tag{12}$$

$$= q_{a_l,i} - \pi_\theta(a_i|s) \tag{13}$$

By applying the chain rule, we can use the above to show that,

$$\left\| \nabla_{\phi} \left( \sum_{j=1}^{k} q_{a_l,j} \log \pi_\theta(a_j|s) \right) \right\| = \left\| \sum_{j=1}^{k} \frac{\partial}{\partial b_j} \left( q_{a_l,j} \log \pi_\theta(a_j|s) \right) \frac{\partial b_j}{\partial \phi} \right\| \tag{14}$$

$$= \left\| \sum_{j=1}^{k} \left( q_{a_l,j} - \pi_\theta(a_j|s) \right) \nabla_\phi w_j^\top f_\theta(\mathbf{s}) \right\| \tag{15}$$

$$\overset{(a)}{\leq} \sum_{j=1}^{k} |q_{a_l,j} - \pi_\theta(a_j|s)|\, l, \tag{16}$$

where (a) follows by the assumption that $\|\nabla_\phi w_j^\top f_\phi(s)\| \leq l$.

$$\left\| \nabla_{w_i} \left( \sum_{j=1}^{k} q_{a_l,j} \log \pi_\theta(a_j|s) \right) \right\| = \left\| \sum_{j=1}^{k} \frac{\partial}{\partial b_j} \left( q_{a_l,j} \log \pi_\theta(a_j|s) \right) \frac{\partial b_j}{\partial w_i} \right\| \tag{17}$$

$$= \left\| \sum_{j=1}^{k} \left( q_{a_l,j} - \pi_\theta(a_j|s) \right) \frac{\partial}{\partial w_i} w_j^\top f_\phi(\mathbf{s}) \right\| \tag{18}$$

$$\overset{(b)}{\leq} |q_{a_l,i} - \pi_\theta(a_i|s)|\, \|f_\phi(s)\| \tag{19}$$

Now, the norm of the gradient of the cross-entropy loss, $\|\nabla \mathcal{L}_{\text{PG-CE}}(\boldsymbol{\theta}, \boldsymbol{s}, \boldsymbol{a})\|$, can be expressed as,

$$\left\| \nabla_\theta A_{\pi_\theta}(s,a) \left( \sum_{j=1}^{k} q_{a_l,j} \log p_{a_l,j} \right) \right\| \leq \|A_{\pi_\theta}(s,a)\| \sum_{j=1}^{k} \left\| \nabla_{w_i} \left( \sum_{j=1}^{k} q_{a_l,j} \log p_{a_l,j} \right) \right\|$$

$$+ \|A_{\pi_\theta}(s,a)\| \left\| \nabla_\phi \left( \sum_{j=1}^{k} q_{a_l,j} \log p_{a_l,j} \right) \right\|$$

$$\overset{(c)}{\leq} \|A_{\pi_\theta}(s,a)\| \left( \sum_{j=1}^{k} |q_{a_l,j} - \pi_\theta(a_j|s)|(\|f_\theta(s)\| + l) \right)$$

Here (c) follows directly by adding inequalities (a) and (b) and completes the proof.

$\square$

## B  PROOF OF PROPOSITION 2

**Proposition 2.** *The norm of $\nabla_\theta \mathcal{L}_{\text{PG-LS}}$ is upper-bounded as follows:*

$$\|\nabla_\theta \mathcal{L}_{\text{PG-LS}}(\boldsymbol{\theta}, \boldsymbol{s}, \boldsymbol{a})\| \leq \frac{1}{\sigma} \left| A_{\pi_\theta}(\boldsymbol{s}, \boldsymbol{a}) \right| \left( l + \|f_\phi(\boldsymbol{s})\| \right) \left| \boldsymbol{a} - \boldsymbol{w}^\top f_\phi(\boldsymbol{s}) \right| \tag{8}$$

*Proof.* We represent $\boldsymbol{\theta} = [\boldsymbol{w}, \boldsymbol{\phi}]^T$ where $\boldsymbol{\phi}$ is network parameters up to and including the penultimate layer and $\boldsymbol{w}$ is the weights of the final linear layer. Note that the mean of the Gaussian policy at state $\boldsymbol{s}$ is $\boldsymbol{w}^\top f_\phi(\boldsymbol{s})$. We start with the Gaussian policy policy gradient surrogate loss:

$$\mathcal{L}_{\text{PG-LS}}(\boldsymbol{\theta}, \boldsymbol{s}, \boldsymbol{a}) = A_{\pi_\theta}(\boldsymbol{s}, \boldsymbol{a}) \frac{1}{2\sigma} (\boldsymbol{a} - \boldsymbol{w}^\top f_\phi(\boldsymbol{s}))^2. \tag{20}$$

We then differentiate this loss w.r.t. $\boldsymbol{w}$ and bound the norm of the resulting gradient:

$$\nabla_{\boldsymbol{w}} \mathcal{L}_{\text{PG-LS}}(\boldsymbol{\theta}, \boldsymbol{s}, \boldsymbol{a}) = A_{\pi_\theta}(\boldsymbol{s}, \boldsymbol{a}) \frac{1}{\sigma} (\boldsymbol{a} - \boldsymbol{w}^\top f_\phi(\boldsymbol{s})) f_\phi(\boldsymbol{s}) \tag{21}$$

$$\|\nabla_{\boldsymbol{w}} \mathcal{L}_{\text{PG-LS}}(\boldsymbol{\theta}, \boldsymbol{s}, \boldsymbol{a})\| \leq \|A_{\pi_\theta}(\boldsymbol{s}, \boldsymbol{a}) \frac{1}{\sigma} (\boldsymbol{a} - \boldsymbol{w}^\top f_\phi(\boldsymbol{s}))\| \|f_\phi(\boldsymbol{s})\| \tag{22}$$

$$\leq \frac{|A_{\pi_\theta}(\boldsymbol{s}, \boldsymbol{a})|}{\sigma} |\boldsymbol{a} - \boldsymbol{w}^\top f_\phi(\boldsymbol{s})| \|f_\phi(\boldsymbol{s})\| \tag{23}$$

We then differentiate the loss w.r.t. $\phi$ and bound the norm of the resulting gradient:

$$\nabla_\phi \mathcal{L}_{\mathrm{PG-LS}}(\boldsymbol{\theta}, \boldsymbol{s}, \boldsymbol{a}) = A_{\pi_\theta}(\boldsymbol{s}, \boldsymbol{a}) \frac{1}{\sigma}(\boldsymbol{a} - \boldsymbol{w}^\top f_\phi(\boldsymbol{s})) \nabla_\phi \boldsymbol{w}^\top f_\phi(\boldsymbol{s}) \tag{24}$$

$$\|\nabla_\phi \mathcal{L}_{\mathrm{PG-LS}}(\boldsymbol{\theta}, \boldsymbol{s}, \boldsymbol{a})\| \leq \|A_{\pi_\theta}(\boldsymbol{s}, \boldsymbol{a}) \frac{1}{\sigma}(\boldsymbol{a} - \boldsymbol{w}^\top f_\phi(\boldsymbol{s}))\| \|\nabla_\phi \boldsymbol{w}^\top f_\phi(\boldsymbol{s})\| \tag{25}$$

$$\leq \frac{|A_{\pi_\theta}(\boldsymbol{s}, \boldsymbol{a})|}{\sigma} |\boldsymbol{a} - \boldsymbol{w}^\top f_\phi(\boldsymbol{s})| \|\nabla_\phi \boldsymbol{w}^\top f_\phi(\boldsymbol{s})\| \tag{26}$$

$$\leq \frac{|A_{\pi_\theta}(\boldsymbol{s}, \boldsymbol{a})|}{\sigma} |\boldsymbol{a} - \boldsymbol{w}^\top f_\phi(\boldsymbol{s})| \cdot l \tag{27}$$

Finally, we combine the inequalities in (23) and (27) to obtain the bound:

$$\|\nabla_\theta \mathcal{L}_{\mathrm{PG-LS}}(\boldsymbol{\theta}, \boldsymbol{s}, \boldsymbol{a})\| \leq \frac{1}{\sigma} \left| A_{\pi_\theta}(\boldsymbol{s}, \boldsymbol{a}) \right| \left( l + \|f_\phi(\boldsymbol{s})\| \right) \left| \boldsymbol{a} - \boldsymbol{w}^\top f_\phi(\boldsymbol{s}) \right|. \tag{28}$$

This completes the proof.

$\square$

## C  EMPIRICAL DETAILS

In this section, we provide additional details about the experiments that were deferred from the main section.

### C.1  ENVIRONMENT DETAILS

In this section, we provide details of the evaluated environments.

1. **Continuous Bandit:** This domain has a single state and the reward is an unknown quadratic function of a $d$-dimensional action. The range for possible actions in each dimension is $[a_{\mathtt{min}}, a_{\mathtt{max}}]$.

2. **Linear quadratic regulator:** LQR is a fundamental control problem in control theory. In this domain, the transition dynamics are a linear function of the state and action with Gaussian noise added. The reward is a quadratic function of the state and action. The action space is 2 dimensional where each dimension is bounded between $[-1, 1]$.

3. **Continous Mountain Car:** In this domain, a toy car attempts to reach the top of a mountain. The action space is 1 dimensional and is bounded between $[-1, 1]$.

4. **Continuous Acrobot:** In this domain, an agent attempts to swing itself above a certain height. The action space is 1 dimensional.

5. **Reacher:** In this domain, a robotic arm tries to reach a goal location. The action space is 2 dimensional and each dimension is bounded between $[-1, 1]$.

6. **HalfCheetah:** In this domain, a cheetah-like robotic agent attempts to run as fast as possible. The action space is 6 dimensional and each dimension is bounded between $[-1, 1]$.

7. **Ant:** In this domain, an ant-like robotic agent attempts to run as fast as possible. The action space is 8 dimensional and each dimension is bounded between $[-1, 1]$.

### C.2  TRAINING DETAILS

The base algorithm is an on-policy actor-critic algorithm and the actual code was based on Stable-Baselines3's implementation of the A2C algorithm (A2C itself is quite similar to the actor-critic algorithm described in Sutton and Barto (2018)) (Raffin et al., 2021). Since it is on-policy, there is no replay buffer. The critic is learned by regressing to observed returns.

For the continous bandit environment, all the algorithms (Gaussian regression, softmax + 1-hot, softmax + HL-Gauss) used a linear policy, used a batch size of 5, and all used a value function baseline. Each algorithm was trained for 2000 interaction steps and the policy was evaluated every interaction step. We tuned the following hyperparameters. For all algorithms, we swept over the following values for the learning rate: $\{10^{-2}, 5 \cdot 10^{-2}, 10^{-1}, 1\}$. For the two classification methods,

we swept over the following number of bins $(k)$: $\{50, 100, 200\}$. For softmax + HL-Gauss, we swept over the following width multipliers $(\eta)$: $\{0.1, 0.25, 0.5, 0.75, 1\}$.

For all the other environments, all the algorithms used the default neural network policy in stable-baseline3 (Raffin et al., 2021), used a batch size of 20, and all used a value function baseline. On LQR, Continuous MountainCar, Reacher, and Acrobot, all algorithms were trained for 500K interaction steps and the policy was evaluated every 500 steps. On the HalfCheetah and Ant domain, all algorithms were trained for 1M interaction steps and the policy was evaluated every 1000K steps. For all these domains and algorithms, we swept over the following learning rate values: $\{10^{-4}, 3 \cdot 10^{-4}, 7 \cdot 10^{-4}, 10^{-3}\}$. For the two classification methods, we swept over the following number of bins $(k)$: $\{50, 100, 200\}$. For softmax + HL-Gauss, we swept over the following width multipliers $(\eta)$: $\{0.01, 0.05, 0.1, 0.25, 0.5, 0.75\}$.

For all algorithms, our hyperparameter sweep is based on picking the hyperparameter combination that led to the highest average undiscounted return (averaged across all trials) on the final step.

### C.3    Hyperparameter Sensitivity Experiments

In this section, we show the sensitivity of the softmax + HL-Gauss algorithm when varying the learning rate, number of bins $(k)$, and width mutliplier $(\eta)$. We conduct the analysis over the values discussed in Appendix C.2.

The sensitivity study is based on holding one hyperparameter fixed (e.g. learning rate) and averaging the returns across all other variable hyperparameters. As such, the returns in the sensitivity studies will always be worse (since high performing and low performing returns may get averaged together) than the one in the main text which shows the highest possible return by a unique combination of hyperparameters. We present the results in Figure 4.

From Figure 4, we find that softmax + HL-Gauss is sensitive to the learning rate, and performance can widely differ based on the value of the learning rate. It is also sensitive to $\eta$, which determines the standard deviation of the histogram Gaussian distribution, where if the standard deviation is too large (larger $\eta$), performance tends to degrade, which is expected since the agent has a challenging time to converge to the optimal actions. The method is quite robust to number of bins $(k)$, where performance is generally stable across all $k$ values.

With regards to the baselines, we see in Figure 5 and Figure 6 that the Gaussian regression method and softmax with 1-hot histogram loss are also sensitive to the learning rates. And softmax 1-hot is similarly robust to number of bins as softmax + HL-Gauss.

### C.4    Performance Profiles

In this section, we report the performance profiles of each algorithm across *all* trials and hyperparameters in Figure 7. These plots illustrate what fraction of the total runs of an algorithm led to a return of greater than $\tau$. In general, we find that the classification losses have a higher fraction of the runs that achieve a high return than the regression loss.

### C.5    Evaluation Returns

In this section, we report (Figure 8) the undiscounted return achieved by the deterministic policy as a function of environment interaction steps.

### C.6    Remaining Training Return Result

Due to space limitations, we include the training returns achieved in Acrobot in this section. This result is part of the results shown in Figure 2.

### C.7    Assets and Software

We implement a standard actor-critic algorithm with the three loss functions using the stabelebaselines-3 framework (Raffin et al., 2021), which uses pytorch for auto-differentiation (Paszke et al., 2019). The HL-Gauss code was built upon the code by Farebrother et al. (2024).

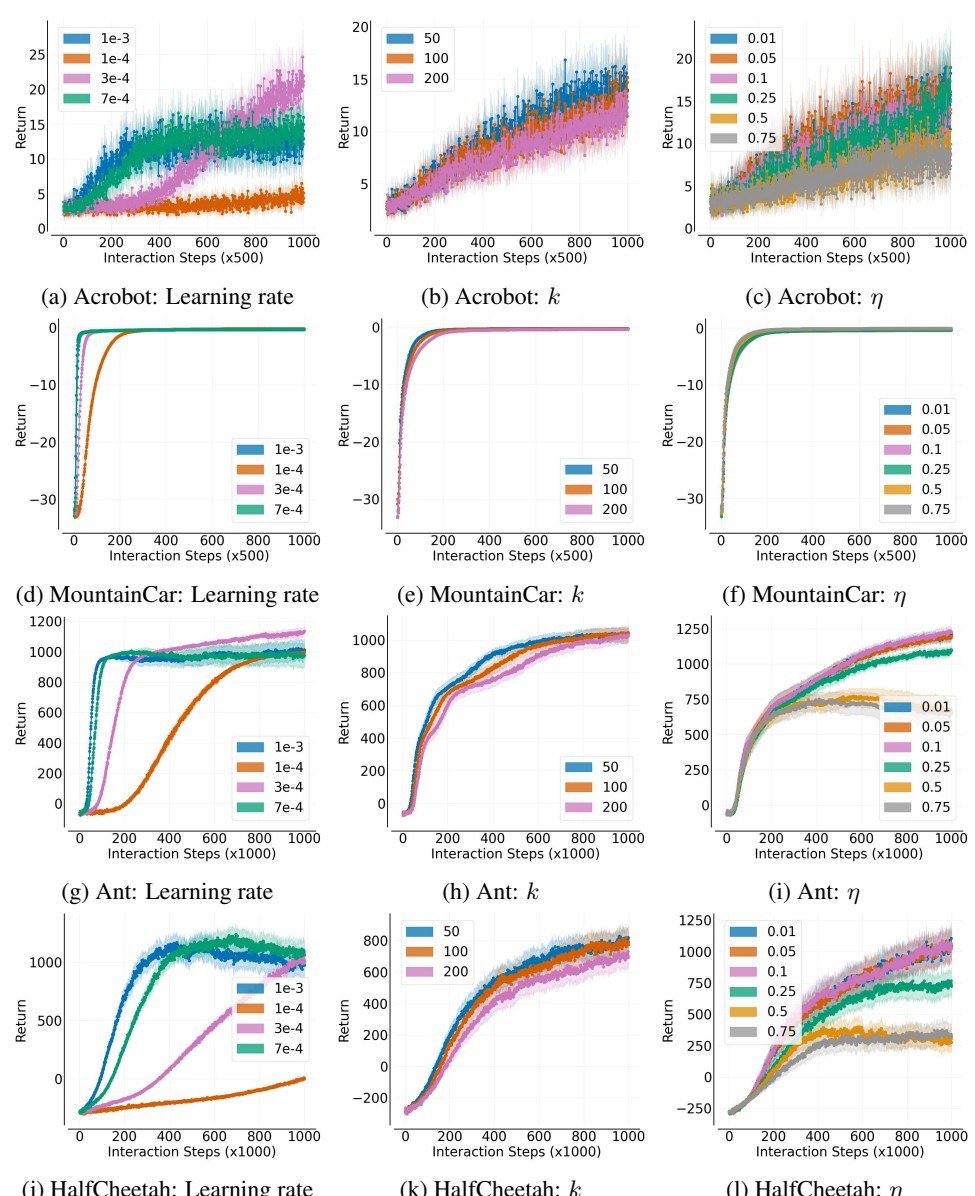

Figure 4: Hyperparameter sensitivity of softmax + HL-Gauss. Undiscounted training returns achieved by softmax + HL-Gauss as a function of environment interaction steps for different hyperparameters. Results are averaged over 20 trials and the shaded region represents the 95% confidence interval. Higher is better.

All the environments were implemented within the gymnasium framework (Towers et al., 2023). We use the rlliable code for plotting (Agarwal et al., 2021).

## C.8 HARDWARE FOR EXPERIMENTS

For all experiments, we used the following compute infrastructure:

- Distributed cluster on HTCondor framework
- Intel(R) Xeon(R) CPU E5-2470 0 @ 2.30GHz
- RAM: 5GB
- Disk space: 5GB

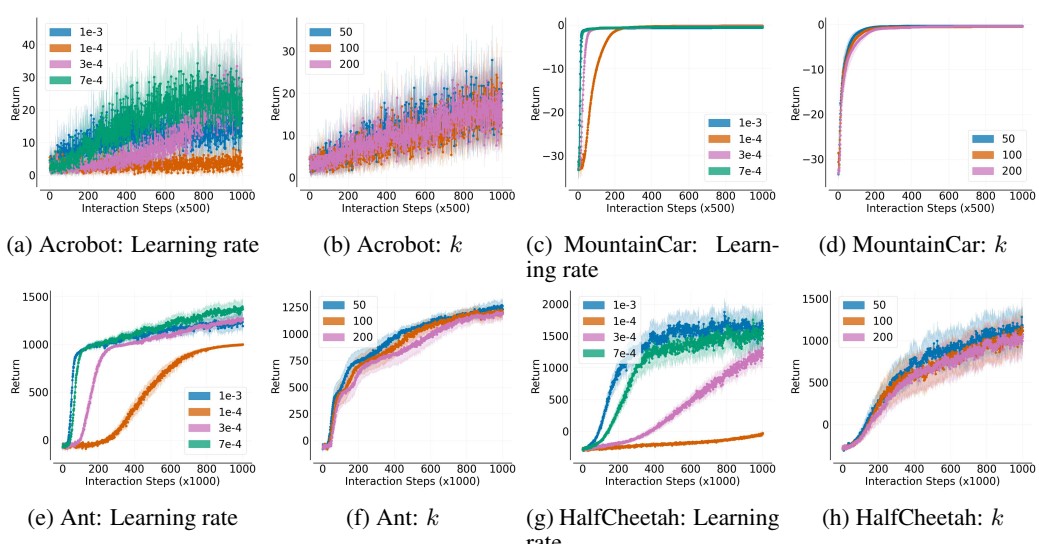

Figure 5: Hyperparameter sensitivity of softmax + HL-1-hot. Undiscounted training returns achieved by softmax + HL-Gauss as a function of environment interaction steps for different hyperparameters. Results are averaged over 20 trials and the shaded region represents the $95\%$ confidence interval. Higher is better.

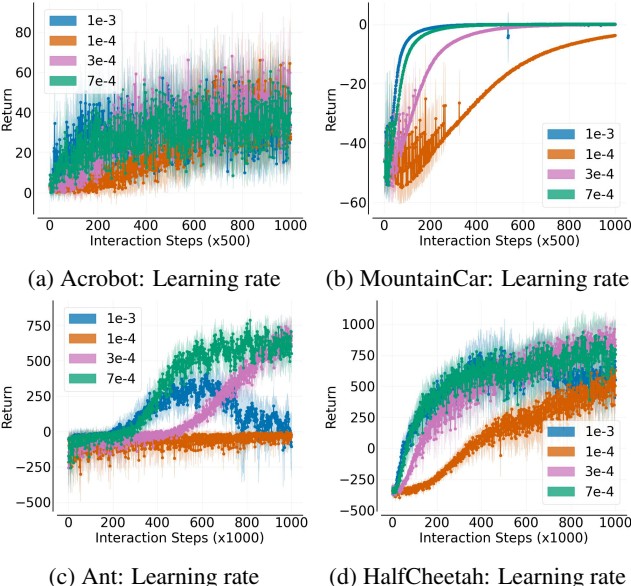

Figure 6: Hyperparameter sensitivity of the Gaussion regression method. Undiscounted training returns achieved by softmax + HL-Gauss as a function of environment interaction steps for different hyperparameters. Results are averaged over 20 trials and the shaded region represents the $95\%$ confidence interval. Higher is better.

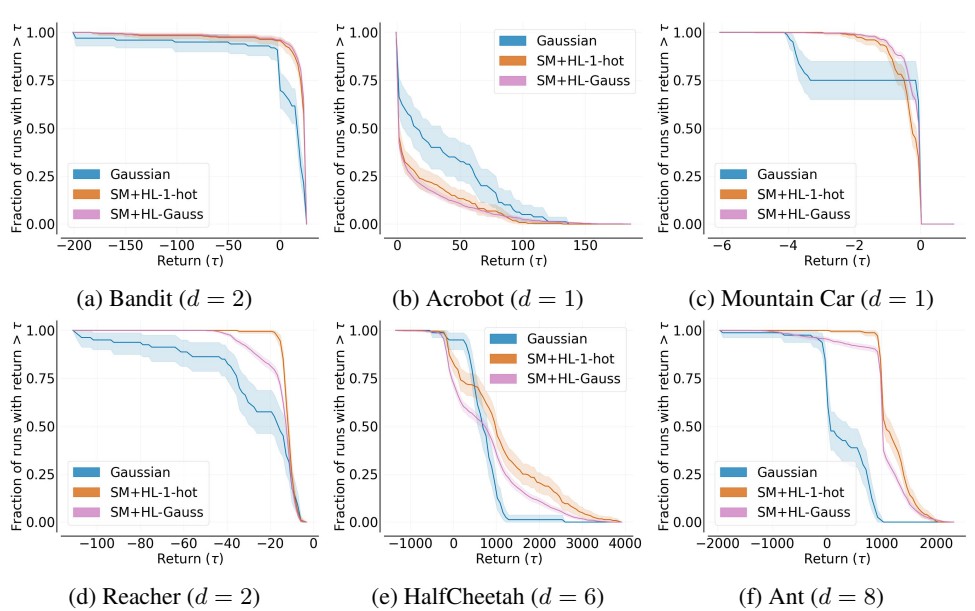

Figure 7: Performance profiles of all the algorithms across all 20 trials and hyperparameter combinations. SM is a softmax policy and HL is the histogram loss. Higher is better. For each domain, we also give the action-dimensionality, $d$.

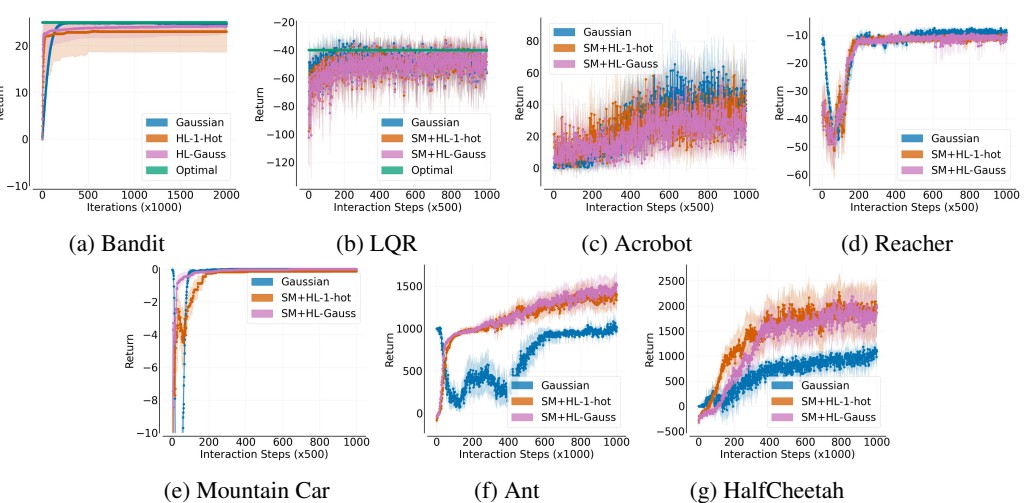

Figure 8: Highest undiscounted *evaluation* returns achieved by each algorithm as a function of environment interaction steps after a hyperparameter sweep. Results are averaged over 20 trials and the shaded region represents the 95% confidence interval. Higher is better. The optimal return can be computed exactly in the Bandit and LQR settings.

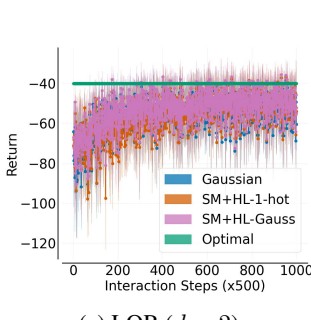

(a) LQR ($d = 2$)

Figure 9: Highest undiscounted training returns achieved by each algorithm as a function of environment interaction steps after a hyperparameter sweep. SM is a softmax policy and HL is the histogram loss. Results are the mean averaged over 20 trials and the shaded region represents the 95% confidence interval. Higher is better.

