# OpenReview forum: "Replacing Implicit Regression with Classification in Policy Gradient Reinforcement Learning"
_ICLR.cc/2025/Conference — Submitted to ICLR 2025_

### Official Review · Reviewer_15B3 · 2024-10-31

**Soundness:** 3
**Presentation:** 2
**Contribution:** 3
**Rating:** 8
**Confidence:** 3

**Summary:**

The paper presents an innovative approach to enhancing the efficiency of policy gradient reinforcement learning by reformulating the implicit regression typically used with Gaussian policies commonly used in continous control as a classification problem. The authors introduce a novel surrogate loss, leveraging cross-entropy loss and softmax policies over discretized actions, and provide both theoretical analysis and empirical evidence supporting this new approach. Overall, the paper addresses a relevant challenge in reinforcement learning, with convincing results and a clear methodology.

**Strengths:**

*Theory*: The paper's proposal to replace regression with classification in policy gradient algorithms is compelling, backed by theoretical bounds showing reduced gradient norms for the cross-entropy-based loss. This theoretical contribution offers insights into the policy optimization algorithms in continous control.

*Experiments*: The paper presents comprehensive experimental results across several continuous control environments, effectively showcasing the advantages of the proposed classification-based surrogate loss. The consistency of the performance gains in data efficiency, stability, and convergence across diverse tasks solidifies the approach's applicability.

**Weaknesses:**

1. The notation $ l $, used initially to represent the Lipschitz constant of the policy network’s output, is later redefined as an index number (line 194). This dual usage is confusing and could benefit from more consistent notation.

2. Certain variables, like $c$, could be recalled where they are used for better readability. For instance, $c$ (defined in line 196) reappears in line 233, but its earlier definition is difficult to locate due to its inline placement in the formula.

**Questions:**

The author assumes that the policy network’s output is $l$-Lipschitz, which seems to be a strong assumption. Could you give some insights why the assumption holds in the experiments?

---

> ### Author Response · Authors · 2024-11-18
> **Author Response**
>
> Thanks for your kind comments and excitement about our paper! We can easily address the weaknesses raised in the camera-ready.
>
> We agree that it may be hard to make an assumption of l-Lipshitz on the output of a network in practice and we do not know for sure that the assumption holds in our experiments. However, we do find that it is easier to tune the step size for the classification methods which aligns with the prediction made by the theoretical results.

---

> ### Comment · Reviewer_15B3 · 2024-11-24
>
> Thanks for the response. $L$-Lipschitz is a rather strong assumption that needs at least some intuitive verifications. It is rather weak to say, "do not know for sure that the assumption holds in our experiments". However, I think the Lipschitz condition will hold for a neural network. The only concern is that the constant may be large, so the whole theorem may be trivial. It will be better to add several numerical experiments to calculate the constant $L$ numerically.

---

### Official Review · Reviewer_SxD3 · 2024-11-02

**Soundness:** 3
**Presentation:** 3
**Contribution:** 2
**Rating:** 5
**Confidence:** 4

**Summary:**

This paper investigates whether replacing implicit regression in policy gradient can improve the training efficiency of policy learning. It introduces a surrogate loss used to reformulate the implicit regression of continuous actions as a classification of discrete actions.  They empirically investigate the use of cross-entropy in the introduced loss as an alternative to the Gaussian policies

**Strengths:**

The paper is well-written and effectively discusses its connections with other methods in the literature, explaining the motivations behind its approach. It provides a thorough theoretical analysis of gradient norms, showing why the classification-based surrogate loss might offer advantages. Additionally, experiments on environments (such as continuous Mountain Car, HalfCheetah, and Ant) demonstrate that the classification-based approach generally improves data efficiency. The paper also includes thorough sensitivity and exploration ablation studies, along with well-documented hyperparameter tuning.

**Weaknesses:**

Discretizing action spaces may lead to performance limitations in domains that require precise control or have high dimensionality, which the authors acknowledge.
The contributions appear incremental, as the method builds heavily on prior work, particularly Imani and White (2018). The term "novel" used in the text may be an overstatement, as the core methodology closely follows established ideas.
The analysis focuses primarily on deriving bounds for gradient norms, suggesting that a smaller bound should lead to greater stability. However, this relationship is indirect, and the analysis does not rigorously connect the smaller gradient norms to concrete improvements in convergence rate. Additionally, key assumptions underlying the theoretical results are not clearly and explicitly stated, making it difficult for readers to assess the validity of the analysis. Furthermore, the propositions are not self-contained or fully descriptive

Minor weakness: The paper lacks clarity in section 5.2.1. It is unclear whether the 20 trials use different seeds, which is important for interpreting the robustness of the results.

**Questions:**

Did you run 20 trials with different seeds, or were the seeds the same for each trial?

I expected the method to perform poorly in higher-dimensional settings, yet the experiments show the opposite. Is there a theoretical justification for this result?

I will increase my score if you address my concerns in the weaknesses part.

---

> ### Author Response · Authors · 2024-11-18
> **Author Response**
>
> Thanks for the kind comments and openness to raising your score. We are happy to discuss further if our comments raise any new questions for you!
>
> Weaknesses:
> 1. Discretization: While these limitations exist in some domains, they will not exist in all domains. Thus, our findings are still valuable to the community given that we have properly acknowledged the limitations. Please note that the dimensionality limitation is more a computational one (number of output units grow linearly) as opposed to an RL-performance limitation, as you noted in our empirical results.
> 2. Novelty: to the best of our knowledge, no prior work has asked or answered the questions that motivate our study. While the work of Ehsan et al. and Farebrother et al. inspired these questions, the answers are not readily available from their works.
> 3. Indirect analysis: we agree with this comment on the theoretical analysis. There is more theoretical work to be done here. Nonetheless, taken together the theoretical analysis and experiments offer valuable insights for the RL community on the choice of loss function in policy gradient learning.
> 4. Theory assumptions: We can easily address this concern in a camera-ready revision. Could you please point out which key assumptions are missing?
> 5. Proposition completeness: We can make the proposition statements more concrete in the camera-ready revision. Just to make sure we understand the concern, is it just to make sure that all notation appearing in the statement is defined there as well?
>
> Questions:
> 1. If we understand this question correctly, we use a different random number to seed each trial. Experiments are ran on CPU so if we used the same seed for all trials then they would all produce identical results. As is best practice, we use common random numbers to reduce variance in the analysis.
> 2. The variance of the gradient estimate for Gaussian policies will increase with dimensionality so we explain this finding as that there is more room for improvement in higher dimensional problems. The only domain where we can do a controlled study of dimensionality is the Bandit domain. There we find that performance does degrade with higher dimensionality though Gaussian policies lose performance more quickly as the dimensionality grows.

---

> > ### Comment · Reviewer_SxD3 · 2024-11-27
> >
> > Thank you for your response. My questions are resolved. Regarding my concerns about the weaknesses, I still believe that the novelty is an overestimation for this paper because the connection between regression and classification has already been developed in the work Imani and White (2018), Farebrother et al. (2024), and Imani et al., (2024). Also, the connection between policy optimization and supervised learning has been previously made by Peters and Schaal (2007); Peng et al. (2019); Abdolmaleki et al. (2018) with a small difference that the authors acknowledged.
> > The novel term is overstated in the sentence that the authors mentioned: "We introduce a novel policy gradient surrogate loss that re-casts the implicit regression of continuous actions as classification of discrete actions." It is a new surrogate loss built heavily on prior work ideas. Moreover, there is indirect analysis which can be more complete.
> > About the assumptions: The l-Lipschitzness of the policy is an important (and common) assumption mentioned in lines 259 and 275, as well as the assumption made in line 186 about independently selecting action dimension. However, they should be stated more clearly and explicitly as an Assumption block using LaTeX for better readability.
> > About the Proposition completeness: Yes.
> >
> >
> > Overall, I decided to keep my decision for this paper unchanged.

---

### Official Review · Reviewer_xobR · 2024-11-02

**Soundness:** 3
**Presentation:** 3
**Contribution:** 2
**Rating:** 6
**Confidence:** 3

**Summary:**

The study introduces an innovative approach to the policy gradient (PG) objective utilized in Gaussian-based policy models within the field of reinforcement learning (RL). It redefines the conventional PG objective by framing it as a weighted squared loss function. In this formulation, the squared loss measures the discrepancy between the chosen action and the action prescribed by the policy, while the weighting factor is derived from the advantage measure, augmented by an additive constant. This conceptual reconfiguration enables the authors to devise a new surrogate function that transitions the methodology from a regression-based to a classification-based model. The paper substantiates the practical utility and enhanced performance of these proposed methods through a series of empirical validations.

**Strengths:**

1. The paper is intellectually interesting, as it proposes some novel angle to think about continuous control based on Gaussian distribution;
2. The presentation is clear;
3. The empirical results look promising.

**Weaknesses:**

I do not have many comments for this paper; so I will keep it short. My primary concerns are below.

1. The reason for why softmax could be beneficial is not clear. Softmax, similar to sigmoid function, could have saturated gradient issue. This has been pointed out by existing work. The author cited one by Shivam et al. in the conclusion, but note that this issue is not restricted to nonstationary learning setting.

2. The paper present some theoretical argument regarding bounded gradient norm, I doubt if it really supports the claim of improved sample efficiency. Note that a strong gradient signal could intuitively improve learning efficiency.

Note that the cited paper by Ehsan et al about histogram loss, they claim improved generalization, rather than sample efficiency.

3. Given the popularity of the deterministic policy, it might be interesting to see how the proposed method compares against deterministic control. Though this is not necessary to support the main claims of this paper.

**Questions:**

see above.

---

> ### Author Response · Authors · 2024-11-18
> **Author Response**
>
> Thank you for your kind words. We’re particularly glad to hear you found the paper to be intellectually interesting! We hope that our answers below will give you a reason to increase your rating.
>
> 1. We agree it is still possible to have a gradient saturation problem but we have not found this to be a challenge in practice as long as the policy is initialized properly. Note that this aligns with results that we have seen in the literature as well (e.g., see Fig 3 of Shivam Garg et al). We would be happy to comment on other references.
>
> 2. We would like to clarify that the theoretical result is intended to support the claim that it is easier to set a step size for learning (see the start of Section 4.3). We also acknowledge Ehsan and White’s argument on improved generalization. The theoretical result itself does not directly support improved data efficiency. If we made a comment that was misleading in this regard, please point it out.
> - There is an indirect benefit to data efficiency in that it is easier to find a step size that leads to data-efficient learning when the gradient magnitude does not change drastically during learning. The bandit domain illustrates this point. Even though the domain is very simple, we found it almost impossible to make Gaussian policies competitive in sample efficiency as any constant step size led to either 1) slow initial learning or 2) instability closer to convergence. We ended up resorting to gradient normalization to make Gaussian policies competitive in the domain.
> 3. Studying deterministic policies is an interesting direction and we think it is possible to also cast DPG as a regression method and ask similar questions. We leave this to future work as it doesn’t help answer our main question here.
>
> We hope that with these clarifications, you will consider increasing your rating and we are happy to provide more discussion if necessary.

---

> > ### Comment · Reviewer_xobR · 2024-12-03
> >
> > Thank you for the rebuttal. The argument about the gradient norm makes some sense. To be fair, I improved my score.

---

### Official Review · Reviewer_RL4Q · 2024-11-05

**Soundness:** 2
**Presentation:** 2
**Contribution:** 1
**Rating:** 3
**Confidence:** 4

**Summary:**

The recent work by Farebrother et al  (2024) showed that instead of training value functions by regression, discretizing the space and using classification instead can yield gains in terms of scalability and performance. Following along these lines, the current work aims to perform something similar for policy-based reinforcement learning. In particular, if we consider a Gaussian policy, the policy gradient w.r.t. $\mu$ is obtained by differentiating a surrogate loss that roughly looks like $A(x-\mu)^2$, which seems like a weighted regression loss (if we consider having target samples at all $x$). Along these lines, the authors change the objective to a weighted classification loss by discretizing the space, and swapping out the $(x-\mu)^2$ bit with a cross-entropy loss.
They performed experiments on applying this idea together with the A2C algorithm on continuous control tasks such as acrobot, mountain car, halfcheetah and ant.
The work also included some theoretical derivations showing that the gradient magnitude is smaller compared to a Gaussian policy.
There were also some other technical additions, e.g., the discretization is performed coordinate wise separately for each action dimension, the logits in each bit affect the probability in nearby bins as well etc. The performance on HalfCheetah went up to around ~2000 (on the Gymnasium version).

**Strengths:**

- The recent results about critic learning with classification are intriguing, so it is interesting to think about whether they are also applicable to policy-based learning.
- Related work appears adequately discussed.

**Weaknesses:**

- The performance of the proposed method on HalfCheetah is only around ~2000, whereas typical good performance on the Gym version is over 10000. Performance on other tasks like Acrobot and Ant, also does not seem close to competitive with good performance on these tasks. Moreover, whilst in the current paper, the A2C benchmark achieves around ~1000 on HalfCheetah, there exist other implementations (e.g., https://tianshou.org/en/v0.4.8/tutorials/benchmark.html) where the plain A2C also achieves ~2000, similar to the proposed method in the current paper. Furthermore, e.g., in Ant the performance of the proposed algorithm in the current paper is ~1500, whereas the tianshou benchmark A2C gets over 3000. From this point of view, the result is not convincing both in terms of whether it really improves over A2C, and also in terms of whether it would improve performance on algorithms that achieve more competitive performance, e.g., PPO.
- I was not convinced by the theoretical results. These results showed that the gradient magnitudes become smaller; however, gradient magnitudes themselves may not tell us about the optimization difficulty. Even just rescaling the objective function will change the gradient magnitudes, but will not change the optimization problem. Perhaps some other metric like the condition number, etc. would have been a more convincing theoretical result to me.
- I didn’t fully see why the method is described as a weighted classification instead of just saying that you are using a discrete policy representation and computing the policy gradient with your representation. The cross entropy is basically the log of the picked actions, but this is the same as in the policy gradient, so couldn’t the method just be seen as switching the Gaussian probability distribution to a discrete one, and applying the standard policy gradient methods?
- The experimental work is not thorough, and is mainly looking at return curves. Providing other kind of experimental evidence in addition to reward curves would be more convincing.

**Questions:**

Why is the performance not competitive?

Why do you consider the method as classification, instead of just thinking of it as changing the policy representation to a discrete one, and applying policy gradients with this new representation?

Small comment:
Equation 4 requires a stop gradient on $A$. Otherwise it is not an unbiased estimator of equation 1.

---

> ### Author Response · Authors · 2024-11-18
> **Author Response**
>
> Thank you for your review. We have carefully considered each of your concerns and responded below. If your concerns remain, we hope to discuss further.
>
> Performance levels are lower than previously reported numbers
> In our implementations, we omit code-level optimizations so as to focus our analysis on the core research questions that we studied. In Deep RL benchmarks, the best-reported numbers usually require a number of techniques such as observation normalization and multi-threaded training. The Tianshou benchmarking page illustrates this point well, showing that differing implementations result in differing performance levels. See also “Implementation matters in deep policy gradients: a case study on PPO and TRPO” by Engstrom et al. We chose to focus on the core issue of the policy parameterization and loss definition by excluding code-level optimizations. Our goal was to understand how the choice of loss function affected performance in a fundamental type of RL algorithm and not necessarily to produce the highest-performing method.
>
>
> We will emphasize this point in our next revision so that it is clear why numbers may differ from values reported elsewhere.
>
>
> Value of theoretical result?
> A tighter bound on the gradient magnitude means the magnitude will vary less across training which is valuable for setting learning rate parameters. We even observed this empirically where, in the simple bandit domain, it was surprisingly difficult to find a constant step-size that worked well for Gaussian policies throughout training. Simply multiplying the objective by a constant would not have this effect.
>
>
> Why consider this classification?
> We agree that in the case of 1-hot that the method is the same as just discretizing the action space. It’s not clear to us how to derive the HL-Gauss loss from a perspective different than the one we adopted. We refer to both 1-hot and HL-Gauss as classification methods to make the connection to results in supervised learning and (more recently) value-based RL. It’s the connection between standard Gaussian policy gradient and regression that motivated this study in the first place. We also think the connection between policy gradient learning and supervised learning is broadly useful and so wished to emphasize it.
>
>
> Thoroughness of experiments?
> Please note that we show additional results in the appendix. We ablate environment properties, attempt to separate out the effects of exploration vs optimization, results for different hyperparameters, and include performance profiles to assess hyper-parameter robustness. Would moving any of these results to the main text address this concern?
>
>
> Small comment: Good point! We will add in our revision.

---

> > ### Comment · Reviewer_RL4Q · 2024-11-18
> > **Not convinced**
> >
> > Thank you for the response.
> >
> > Regarding the performance levels, Farebrother et al. combined classification with SOTA methods, and achieved SOTA results. The current work, on the other hand, only has low performing results. The apparent improvement in performance that you are seeing could also be specific to the implementations and parameterizations that you picked. Without high-performance results it is not clear whether your suggestions are applicable to well-performing implementations, and it is also not clear whether the result is specific to your implementation or whether it generalizes to other ones. For that reason, I don't find it convincing unless you at least outperform the SOTA algorithms of the same class.
> >
> > Regarding the bound, you only bound the magnitude of the gradient, not it's variance. Also, you mention that "Simply multiplying the objective by a constant would not have this effect.", which is not true. If the gradients vary between -C and C, and I divide the loss function by 2, then they will start varying between -C/2 and C/2. Clearly the variation decreases, but it will not affect the optimization difficulty.
> >
> > To derive the HL-Gauss loss, my understanding is that you should first discretize the state space, say to actions a_1, ... a_N. Then you can parameterize the probability assigned to each action, and you can derive the HL-Gauss method. In particular, you can create some Gaussians centered at a_1, ..., a_N, and when computing the probability assigned to each action, you sum across all of the Gaussians. It seems just like a different way to parameterize the probability assigned to each discrete location.
> >
> > Regarding the experiments in the appendix, I was aware of them, and they only look at reward curves, with the only exception being the performance profiles, which is a different way to visualize the reward results. I would have expected experiments looking at other statistics in addition to rewards. How does the new policy parameterization affect the learning dynamics? Is the action selection more diverse? Why do you think that the performance improves? I would have expected experiments showing evidence for these kinds of points. For example, you gave a theorem about the gradients, but you don't compute or plot the gradient magnitudes or their variances. None of the experiments give a compelling explanation for improved performance.

---

> > > ### Author Response · Authors · 2024-11-18
> > >
> > > Thanks for a quick response to us and engaging with discussion. We believe there are two questions raised by your first point: 1) is it necessary to have SOTA results to advance the RL community’s understanding of policy gradient learning? and 2) can the community have confidence that this paper has done everything possible to set up a fair comparison for this purpose?
> > > 1. We respectfully disagree that SOTA results are necessary for advancing understanding in RL research. In fact, pursuing SOTA by including many different code-level optimizations can even be in opposition to this goal. For example, see Section 5 of https://arxiv.org/pdf/2005.12729, where it is shown that code-level optimizations to PPO have hindered understanding of the true effectiveness of the clipping mechanism. An alternative approach is to use the simplest base algorithm as a means to reduce the confounding impact of other techniques that are not the current object of study.
> > > 2. With the above in mind, we initially tried to conduct our study with as basic an actor-critic algorithm as we could. We began with SB3’s A2C implementation as A2C is already fairly similar to the basic actor-critic algorithm described in “RL: an Introduction.” We confirmed that the implementation could learn with Gaussian policies before formally tuning hyper-parameters or implementing HL-Gauss. Once we were confident in the implementation, we tuned all methods using the same procedure as reported in the appendix. To the best of our ability, the reported numbers represent a fair comparison between a base Gaussian policy actor-critic algorithm and an actor-critic algorithm using either 1-hot or HL-Gauss cross-entropy losses. We also point out that some of our numerical values are close to what other works have reported according to Tianshou’s benchmarking. For example, “Gaussian” matches PPO in the Spinning Up implementation for Ant, is similar to Spinning Up’s TRPO for HalfCheetah, and is similar to PPO from the PPO paper on Reacher.
> > >
> > > We see your point but don’t believe it changes the analysis of our paper. Suppose we multiply each loss by a positive constant that is < 1. That factor would simply appear on the RHS of each proposition and so “cancel” out in the comparative analysis we provide. We can also improve the discussion of the bounds by citing Chapter 9 of “Optimization for Data Analysis” (Wright and Recht 2022) which explains that the sub-optimality gap after a fixed number of optimization steps is upper bounded by a term that depends on an upper bound of the gradient magnitude.
> > >
> > > There might be a misunderstanding about the HL-Gauss method. With HL-Gauss, the policy is still parameterized as a softmax distribution. Between 1-Hot and HL-Gauss, there is no difference in the policy network. The difference is only in the loss function. 1-hot minimizes a weighted cross-entropy between the current policy and a 1-hot distribution on the action taken. HL-Gauss minimizes a weighted cross-entropy between the current policy and a histogram approximation of a Gaussian centered at the action taken. The effect is that 1-hot only only reinforces the action that was taken, whereas HL-Gauss primarily reinforces the action that was taken but also reinforces nearby actions to a lesser extent.
> > >
> > > Additional metrics: thanks for clarifying what you had in mind and we see your point. We’re taking a look at the gradient norms in particular and will report back.

---

> > > > ### Comment · Reviewer_RL4Q · 2024-11-19
> > > >
> > > > >We respectfully disagree that SOTA results are necessary for advancing understanding in RL research.
> > > >
> > > > I never said that SOTA results are necessary for advancing understanding in RL research, so there is nothing to disagree on. I would appreciate if you only argue against points that I actually said without distorting my viewpoint.
> > > > There are a spectrum of possible useful contributions from different points of view: either increasing our theoretical understanding, solving new problems, or improving performance on existing tasks. I judged that the current work is weak in terms of theoretical results, increasing our understanding or novelty. For this reason, I would have liked to see stronger empirical performance, but the current work is not strong from that point of view either. I am interested in mainly two questions. Is the work useful or interesting? Is it likely to lead to something useful or interesting? I am not confident in a positive answer to either of these. Certainly the performance right now is not competitive, and I also don't see any reason to suspect that it will transfer to stronger algorithms. There are many different policy distribution parameterizations that could be considered, and I don't see why the current one should be good. There are also other potential issues, such as loss of fine control due to the discretization.
> > > >
> > > > Regarding multiplying the objective function, I think you misunderstood my point. Your argument was roughly: the gradient magnitudes are smaller in our method, therefore, the optimization is better and your method improves. However, my argument is that the second part is incorrect. Smaller gradients are not a sufficient condition for better optimization performance, so your theorem does not tell us anything about the optimization performance. My argument with multiplying the objective function by a constant provides a counter-example to the claim that reducing gradient magnitudes will make optimization easier. By providing a counter-example, I disproved the claim, so your logic is flawed. Regarding convergence proofs that depend on the square of the gradient, these proofs are typically based on optimizing the same objective function with stochastic gradient descent, but with differently computed gradients. In this case, the expected value of the gradient stays the same, but the expectation of the squared gradient may decrease (E[X^2] = E[X]^2 + V[X]) caused by a reduction in the variance of the gradient. However, in your case, changing the policy parameterization changes the objective function, so there is currently no guarantee of improved performance. Perhaps your results could be extended somehow to show better optimization, but the current theoretical result is insufficient to make any claim about optimization speed, or at least there are gaps in the arguments.
> > > >
> > > > Regarding HL-Gauss, perhaps it is indeed not exactly the same, but I believe it is conceptually similar. The logits could be parameterized as y = Az, where A represents the matrix performing the Gaussian weighting. Then, when the logits are updated by gradient descent, the gradient going into y_i, will propagate to other values in z based on the weights, and in turn the weights in A will lead to a change in the other y values when multiplied by z. In practice z would be parameterized by a neural network, but conceptually, it is achieving the same effect of updating nearby logits as well. In either case, it's not an important point. My argument is just that similar effects can be achieved by purely parameterizing the distributions in some specific way.

---

### Meta-Review · Area_Chair_eQYV · 2024-12-21

**Metareview:**

This work provides the point of view that continuous control using policy gradient updates with Gaussian policies is a form of weighted least-squares regression. It leads to a natural conjecture that it may benefit by reformulating as a classification task, which some works have suggested for supervised learning as well as value-based reinforcement learning. Consequently, the authors study whether this is beneficial using A2C an on-policy policy gradient method as a baseline. On mujoco Gym tasks halfcheetah and ant, they indeed show reasonable performance, although far from current competitive results. And the result is undermined by the known state of the art results of A2C on these two tasks.

You noted “in reality Equation (1) is not the gradient of J” but then why write “nabla J” in (1)? And later you claim that the gradient of (4) is an unbiased estimate of (1), which again contradicts your above statement. Also some concise explanation of why it is biased is possible and would be appreciated by the readers.

It is not clear why the experiments were not redone using more commonly used policy gradient methods such as PPO and SAC. Due to many significant shortcomings of this paper, I recommend submitting the paper to a future venue.

**Additional Comments On Reviewer Discussion:**

While the paper is going toward an interesting direction, the shortcomings I mentioned above are also shared by some of the reviewers, resolving which would require substantially more work. Reviewer RL4Q gave extensive arguments why these are necessary for a high-quality empirical conclusion. Future work should address fine control in high-dimensional domains, perform additional ablations such as gradient norm and variance analysis, and explore alternative discretization or policy formulations. The authors would benefit from adopting these suggestions.

---

### Decision · Program_Chairs · 2025-01-22

Reject